# Did people really drink bleach to prevent COVID-19? A guide for protecting survey data against problematic respondents

Leib Litman[1,2]*, Zohn Rosen[3], Rachel Hartman [1,4], Cheskie Rosenzweig[1,5], Sarah L. Weinberger-Litman[6], Aaron J. Moss[1], Jonathan Robinson[1,7]

**1** CloudResearch, New York, New York, United States of America, **2** Department of Psychology, Lander College, New York, New York, United States of America, **3** Department of Health Policy and Management, Mailman School of Public Health, Columbia University, New York, New York, United States of America, **4** Department of Psychology & Neuroscience, University of North Carolina at Chapel Hill, Chapel Hill, North Carolina, United States of America, **5** Department of Clinical Psychology, Columbia University, New York, New York, United States of America, **6** Department of Psychology, Marymount Manhattan College, New York, New York, United States of America, **7** Department of Computer Science, Lander College, New York, New York, United States of America

* leib.litman@touro.edu

**Data Availability Statement:** All data files are available on OSF at https://osf.io/fzx9v/?view_only=90d2039f61384f9b9dd99b72ca547c9a.

## Abstract

Survey respondents who are non-attentive, respond randomly, or misrepresent who they are can impact the outcomes of surveys. Prior findings reported by the CDC have suggested that people engaged in highly dangerous cleaning practices during the COVID-19 pandemic, including ingesting household cleaners such as bleach. In our attempts to replicate the CDC's results, we found that 100% of reported ingestion of household cleaners are made by problematic respondents. Once inattentive, acquiescent, and careless respondents are removed from the sample, we find no evidence that people ingested cleaning products to prevent a COVID-19 infection. These findings have important implications for public health and medical survey research, as well as for best practices for avoiding problematic respondents in all survey research conducted online.

## Introduction

Surveys are one of the most common sources of data in social science [1–3], political science [4–7], public health [8, 9] and medical research [10, 11], informing public policy, medical practice and public opinion. Despite the widespread use of survey research, self-report data has come under increasing scrutiny due to data quality concerns [12–14].

One major threat to validity in survey research comes from participants who are inattentive [12, 15, 16], "mischievous", providing responses that are intentionally false, or systematically responding "yes" to any question [17–23]. In the present research, we examine how these "problematic respondents" can bias the results of surveys by dramatically inflating point estimates and by creating illusory associations in the context of health-related behaviors. Specifically, we examine the validity of a Centers for Disease Control and Prevention (CDC) study

**Funding:** The author(s) received no specific funding for this work.

**Competing interests:** The authors have declared that no competing interests exist.

that found Americans engaged in highly dangerous practices in response to the COVID-19 pandemic, including the ingestion of bleach and household cleaner [24].

Estimates of rare events, such as the ingestion of household cleaning products, are particularly prone to problematic-respondent bias [18, 25]. The goal of the present study is to examine whether the rate of reported dangerous cleaning practices, and the relationship between dangerous cleaning practices and health outcomes, were overinflated in the CDC study [24] due to problematic respondents. Another goal of this paper is to examine approaches to reducing problematic responses and demonstrate how data quality can be vastly improved when collecting data online.

## Inflated estimates

Problematic respondents can alter the outcomes of surveys, as observed in the 1970s in surveys on self-reported illicit drug use. Petzel and colleagues created a paradigm for catching potentially problematic respondents which involved incorporating questions about a fictitious drug in their survey [26]. They found that 4% of people reported using this fictitious drug, and that these people were also much more likely to report using other drugs, suggesting a general propensity toward acquiescence bias.

A nationwide school-based study in Norway used a similar approach, asking close to 12,000 participants questions about drug use [27]. The authors found that respondents who reported buying and using the fictitious drug "Zetacyclin" also reported disproportionately heavy use of other drugs like heroin and LSD. Because heroin and LSD use is rare, excluding these respondents made a critical difference for inferences about nationwide drug use.

The national Longitudinal Study of Adolescent Health (Add Health) [17] has provided some of the most compelling demonstrations of the impact that problematic respondents can have on surveys. Add Health uses multiple measurement methods, including surveys and in-person interviews, which allows for survey responses to be cross-referenced. There have been multiple instances where in-person interviews directly contradicted survey responses. For example, 20% of respondents falsely reported not being born in the US, and 19% falsely reported being adopted. These responses were later contradicted by the adolescents' parents during in-person interviews. In what is perhaps the most striking example of the potential for problematic respondents to invalidate the results of medical surveys, only 2 out of the 253 people who indicated that they had used an artificial limb for more than a year were confirmed to have done so by direct observation in a follow-up in-person interview [17].

## Inflated correlations

Once a respondent provides a false response, they are more likely to provide other false and problematic data. Demonstrably false responses to some questions are typically a good indication that the entire survey should be treated with suspicion and considered for exclusion from the analysis. For example, people who falsely reported being adoptees or having a false limb also endorsed extreme responses on a range of behaviors, leading to inflated and illusory between-groups disparities. A reanalysis of the Add Health data after removing problematic respondents led the authors to conclude that several original reported disparities were "substantially overstated," and led to retractions of published reports [17, 23].

## Mitigating against problematic respondent bias

[15, 28–34] Problematic respondent bias is a ubiquitous problem that requires mitigation in any type of survey [20, 28], regardless of the modality or demographic population. Thus, the inclusion of rigorous methodology to support the validity of estimates drawn from survey data

is critical [35]. To this end, researchers have developed data validity screening instruments to combat problematic responses. These instruments can be added before the survey to prevent problematic respondents from participating [28], or they can appear within the survey to identify problematic respondents to be excluded from the analytic sample [15, 29–34]. [36] Using such instruments has helped reveal that problematic respondents can drastically attenuate results, at times leading researchers to conclude that previously established findings lack validity [17, 36, 37]. This is especially important for studies with direct implications for public health and public policy [13, 20, 28, 35].

## Problematic respondents in online surveys

One popular modality of collecting survey responses is via online opt-in panels, which constitute more that 80% of currently conducted public opinion polls [25], and are increasingly used in public health, political science, and social and behavioral sciences [28, 38]. A large literature on opt-in panels indicates that the percentage of problematic respondents on such panels is substantial [6, 14, 25, 28, 39–44]. Estimates of the magnitude of problematic respondent bias in online opt-in panel platforms vary between 4–7% [25] and 30% [28], although in some studies the magnitude of inattention has been as high as 50% [14].

Problematic survey responses in online panels are not random, and tend to be skewed toward positive answers [25]—i.e., problematic respondents will be more likely to choose "yes" over "no". This acquiescence bias is particularly concerning in studies that aim to measure rare events, because even a small percentage of respondents who falsely answer "yes" to questions about rare events will make a non-existent phenomenon appear to be real. The present study addresses this issue in the public health context. We also explore whether reported correlations between dangerous health behaviors and negative health outcomes may be inflated due to problematic responses.

## Health behaviors during COVID-19

The COVID-19 pandemic impacted daily health-related practices in the United States and around the world, and the World Health Organization (WHO) and CDC issued guidelines to help curb its spread, especially regarding cleanliness practices such as the need to wash hands thoroughly and often, and to avoid hand-to-face contact [45, 46].

Previous research has shown that even before COVID-19, people engaged in various cleaning practices to reduce the likelihood of infection, particularly surrounding food cleanliness [47, 48]. At times, contamination concerns can lead to dangerous cleaning practices, such as overusing antimicrobial products [49]. It is thus reasonable to expect that during a pandemic, when fear of infection is very high, people will engage in even more cleanliness practices to protect their health.

While there is some evidence of an increase in dangerous practices in the reported rise in calls to poison control centers at the start of the pandemic [50], it is unclear if the greater volume of calls was due to intentional or unintentional ingestion/inhalation of cleaning products. The rise in calls to poison control centers could be due to an overall increase in cleaning practices, leading to more accidental exposure to dangerous chemicals. The likelihood that most of these incidents were accidental is corroborated by several observations.

First, half of the calls to poison centers were for children under age 5. Furthermore, while there were 28,158 calls to poison control centers related to disinfectants in January-March of 2020, there were 25,021 such calls during the same period in 2019. This suggests a high base rate of contamination incidents from improper or accidental use of disinfectants, unrelated to COVID-19. These incidents are likely due to accidental contamination by children and adults,

as well as the use of industrial bleaches such as chlorine dioxide by people who falsely believe in their medicinal properties. These beliefs are more common in Latin America and among certain groups, including physicians, who favor homeopathic and non-traditional medicine [51]. This use of industrial bleach for medical purposes precedes COVID-19 by decades [52].

Additionally, several practices examined in this study, including the ingestion and inhalation of household cleaner, have been documented in the medical literature [53]. Such practices are commonly observed among vulnerable populations like low socioeconomic groups, teenagers, and prison inmates. There have been several documented cases of blindness and death from drinking hand sanitizer since the start of the COVID-19 pandemic [54, 55], but in virtually all cases, people ingest and inhale sanitizer for its alcohol content and psychotropic effects [56]. It is possible that the use of sanitizer for its psychotropic effects increased at the start of COVID-19 due to the psychological toll of the pandemic, particularly on vulnerable groups.

Overall, reported disinfectants-related incidents among adults increased by a few thousand at most during the first three months of the COVID-19 pandemic compared to previous years [50]. While this increase is significant, it constitutes less than 0.001% of the US population. However, the CDC proceeded to further investigate the issue.

In June 2020, the CDC reported the results of a survey they conducted using an online opt-in panel [24]. The survey asked American respondents if they had engaged in several cleaning practices to prevent a COVID-19 infection during April 2020. Their data revealed that 39% of Americans engaged in at least one cleaning practice not recommended by the CDC. For our purposes, we categorize these practices as either moderately or highly dangerous. Moderately dangerous practices include washing food products with bleach (19% of respondents), using household cleaner or disinfectant on one's skin (18%), misting the body with cleaning or alcohol spray (10%), and inhaling the vapors of household cleaners like bleach (6%). Highly dangerous practices included drinking or gargling household cleaning products (4%), drinking or gargling soapy water (4%), or drinking or gargling diluted bleach (4%) in order to prevent a COVID-19 infection.

The finding that Americans were engaging in dangerous cleaning practices at high rates is alarming, and appears to align with the reported increase in calls to the CDC poison control center. Taken together, a narrative emerged, suggesting that fears of COVID-19, coupled with a lack of knowledge about the dangers of such practices, were leading tens of millions of people to engage in behaviors that can damage their health.

However, these results should be interpreted with caution due to the potential for problematic responses to bias estimates of such rare behaviors. For this reason, we sought to examine whether reports of dangerous cleaning practices, especially the ingestion of cleaning products, can be attributed to problematic respondents.

## The present research

The goal of this study was to examine whether reports of dangerous cleaning practices, such as ingesting household cleaners to prevent COVID-19 infection, can be detected after controlling for problematic respondent bias. Across two samples collected in June and July of 2020, we measured the magnitude of problematic respondent bias and its influence on estimates of dangerous cleaning practices, with a focus on the ingestion of household cleaners including bleach, soapy water, and household disinfectant. We aimed to determine the role of problematic respondents in reporting these practices and to more accurately measure their prevalence. Since problematic responses may also artificially inflate correlations between measures, we also investigated whether dangerous cleaning practices remained associated with negative health outcomes after removing problematic respondents from the sample. Data and syntax are available at https://osf.io/fzx9v/?view_only=90d2039f61384f9b9dd99b72ca547c9a.

## Hypotheses

H1: Problematic respondents will be responsible for most reports of dangerous cleaning practices, especially the three highly dangerous practices: drinking or gargling bleach, disinfectant, or household cleaner. Conversely, very few non-problematic respondents (less than 1%) will report engaging in these behaviors.

H2: Dangerous behaviors with lower (vs. higher) reported frequencies will have a greater proportion of affirmative responses from problematic respondents.

H3: The response verification procedure will reveal that non-problematic respondents who report ingesting cleaning products either fail to confirm doing so in a follow-up question, or indicate doing so unintentionally, rather than purposely as a COVID-19 preventative measure.

H4: An examination of respondents' open-ended responses will reveal few, if any, cogent descriptions of respondents' ingestion of cleaning products. In contrast, there will be informative open-ended responses of other moderately dangerous practices, such as washing produce with bleach.

H5: There will be a significant correlation between the rates of reported ingestion of household cleaning products and other implausible/impossible behaviors, reflecting an acquiescence bias.

H6: The association between dangerous cleaning practices and negative health outcomes will be high among problematic respondents and low among non-problematic respondents, again reflecting an acquiescence bias.

## Method

### Participants and design

The current study was a replication of a study conducted by the CDC in May 2020 [24]. Aside from the addition of data quality measures, we used the identical survey design, question wording, online sample provider, and sampling methodology as reported by the CDC. The study was exempt from review because it is an anonymous survey. Exempt status was confirmed by IntegReview IRB.

We collected Sample 1 from a national sample of 600 respondents during the week of June 10th-June 17th, 2020, and Sample 2 from a national sample of 688 respondents during the week of July 27th-July 31st, 2020. The samples were matched to the U.S. Census on gender, age, race, and region (see Table 1 for the respondents' demographics). After providing informed consent by clicking "next" at the end of the consent form, participants responded to the measures described below.

### Materials

**Cleaning practices.**   The survey included questions about the cleaning behaviors respondents engaged in as a response to the COVID-19 pandemic. In addition to asking about an increase in housecleaning frequency (which would not be considered dangerous), the moderately dangerous practices included washing produce with bleach, using household cleaner to clean or disinfect one's skin, misting the body with cleaning or alcohol spray, inhaling the vapors of household cleaners, and the highly dangerous practices included drinking or gargling household cleaner, soapy water, or diluted bleach.

**Negative health outcomes.**   Participants indicated whether they had experienced any of a list of health effects due to using cleaners or disinfectants "in the past month" (Sample 1) or "since the start of the COVID-19 pandemic in April" (Sample 2). These include nose or sinus irritation, skin irritation, eye irritation, dizziness, lightheadedness, or headache, upset stomach or nausea, and breathing problems.

**Table 1. Participant demographics.**

| Variable | Sample 1 | Sample 2 |
|---|---|---|
| **Age** | *M* = 50, *SD* = 17.2 | *M* = 43.1, *SD* = 17.8 |
| **Gender** | | |
| **Female** | 53.2% | 55.8% |
| **Male** | 46.3% | 37.9% |
| **Race** | | |
| **White** | 71.7% | 63.1% |
| **Black** | 12.8% | 12.1% |
| **American Indian or Alaska Native** | 2.2% | 2% |
| **Asian** | 7.8% | 7.3% |
| **Native Hawai'ian or Pacific Islander** | 0.2% | 0.9% |
| **Multiracial** | 2.5% | 5.7% |
| **Other** | 2.8% | 3.7% |
| **Ethnicity** | | |
| **Hispanic** | 16.8% | 21.5% |
| **Not Hispanic** | 83.2% | 73% |
| **Highest degree** | | |
| **No college** | 34.7% | 48% |
| **Associate degree** | 10% | 12.5% |
| **Bachelor's degree** | 30.7% | 24.9% |
| **Graduate degree** | 24.7% | 14.5% |
| **Region** | | |
| **Northeast** | 25% | 23.7% |
| **Midwest** | 20.8% | 23% |
| **South** | 35.2% | 30.8% |
| **West** | 19% | 17% |

**Data quality measures.** We employed a combination of instruments to address multiple known characteristics of problematic respondent bias.

*Attentiveness and English language comprehension*. We adapted a procedure developed by Chandler and colleagues [28] to check for attentiveness and basic English language comprehension. Participants saw a target word and indicated the most related word from a list of four other words. This instrument identifies the vast majority of inattentive respondents in online samples, with very low levels of false positives. We refined and extensively tested the instrument by using an associative semantic network algorithm to assign weights to word-pairs based on corpora of English language texts, and using only very common words to avoid education bias. We used screening questions with a pass rate of 95% or above based on pilot testing on high quality samples. This allows different combinations of screening questions to be presented to different participants, preventing bots and problematic respondents from learning correct responses and creating scripts for response automation. This is the same method used to create Sentry, CloudResearch's tool for behaviorally validating online samples.

In addition to this measure, we also included two additional questions in Sample 1. These questions were (1) The trophy doesn't fit into the brown suitcase because it is too large. What is too large?" and (2) "Have you ever used the internet?". Each of these had several response options, with only one correct answer ("The trophy" and "Yes", respectively).

*Acquiescence*. The second method we used was originally developed by Petzel and colleagues [26] and subsequently used by multiple other methodologists [27, 28, 36]. It involves incorporating questions within the survey about highly unlikely or impossible behaviors to

identify problematic respondents. We asked respondents three questions where the only plausible answer is "No." Specifically, in Sample 1, we asked (1) "Do you know what the word wuttlet means?" (2) "Have you ever suffered a fatal heart attack?" and (3) "From memory, can you recall the name of every senator who has ever served in the U.S. Senate?" In Sample 2, we asked "Do you ever eat concrete for its high iron content?" Each of these questions was pretested to ensure that over 95% of attentive respondents answer in the expected way.

*Demographic and response verification (Sample 2 only).* We used Robinson-Cimpians' method of validating demographic information [18] to further identify problematic respondents. This method involves looking across reported demographics to find inconsistencies and exaggerated claims. Robinson-Cimpian developed a quantifiable metric for flagging problematic respondents based on an outlier analysis. Here, we utilized a similar approach by flagging demographic claims that are clearly implausible. Among respondents who passed all previous data quality measures (i.e., found to be attentive, not mischievous, and verified that they had intentionally engaged in dangerous cleaning practices as a preventative COVID-19 measure), we examined their reported age, height, weight, and parental status, looking for extreme outliers and implausible entries.

As an additional verification measure, Loftus et al. showed that one way to increase accuracy in survey responses is to ask respondents to report on behaviors of interest multiple times [57]. Their study revealed that patients tend to overreport whether they have had a physical examination, as verified by patient records [58]. They also found that incorporating a second question to verify the original response significantly improves reporting accuracy. This occurs in part because it signals to participants the importance of the question to the researcher [57].

We used an approach similar to Loftus et al., which consisted of multiple steps. First, after respondents indicated they had engaged in a dangerous cleaning behavior, they received a follow-up question asking whether they intended to respond affirmatively. For example, "You indicated you drank or gargled diluted bleach solution. Did you really drink or gargle diluted bleach solution, or did you indicate you did so by mistake on the last survey question?" Next, respondents who verified that they had indeed intended to respond affirmatively received another follow-up question to verify that the respondent had intentionally engaged in the behavior. For example, "You indicated you drank or gargled diluted bleach solution. Did you engage in this cleaning behavior intentionally?" Finally, we asked all respondents who reported engaging in a dangerous cleaning behavior to provide more context about their answer in an open-ended format: "Please describe the steps you took to clean this way. What cleaning product did you use? How did you administer it? This research is very important for public health policy and we very much appreciate your time and input!".

## Results

We first classified respondents into one of two groups: "problematic respondents" were those who responded incorrectly to any of our data quality measures, and "non-problematic" respondents were those who responded correctly to all data quality measures. In Sample 1, 460 respondents (76.7%) were "non-problematic," and 140 respondents (23.3%) were problematic. In Sample 2, which included additional response verification (see the Response verification section below for more details), 461 respondents (67%) were "non-problematic," and 227 respondents (33%) were problematic.

### Who reports engaging in dangerous cleaning practices?

Across the full samples we found that, absent any data quality controls, the rates of reported cleaning and disinfection practices in our samples closely mirrored those reported by the CDC [24]. See Table 2.

**Table 2. Reported cleaning and disinfection practices across Gharpure et al. (CDC), Sample 1, and Sample 2.**

| Cleaning and disinfection practice | No data cleaning | | |
|---|---|---|---|
| | CDC | S1 | S2 |
| Increased frequency of home cleaning | 60% | 56% | 54% |
| Washed fruits, vegetables, or other food products with bleach | 19% | 12% | 13% |
| Used household cleaner to clean or disinfect hands or bare skin | 18% | 16% | 17% |
| Misted the body with cleaning spray or alcohol spray after being in public spaces | 10% | 12% | 10% |
| Inhaled the vapor of household cleaners like bleach | 6% | 6% | 5% |
| Drank or gargled a household cleaner | 4% | 4% | 4% |
| Drank or gargled soapy water | 4% | 4% | 4% |
| Drank or gargled diluted bleach solution | 4% | 4% | 4% |

To address H1, we examined the reports of engaging in dangerous cleaning practices among problematic vs. non-problematic respondents. As predicted, problematic respondents provided the vast majority of the affirmative responses to questions about dangerous cleaning behaviors, particularly for the highly dangerous behaviors (Fig 1). Most notably, no none-problematic respondents in Sample 2 reported any cleaner ingestion.

The opposite pattern was observed for increases in typical non-dangerous cleaning practices. While it is expected that normal cleaning behaviors would increase during the pandemic, driven by public health recommendations, problematic respondents under-reported increases in normal cleaning behavior by close to 20%. Thus, problematic respondents severely overreport high-risk cleaning practices and underreport regular cleaning practices.

Confirming H2, we found that dangerous behaviors that are less common had a greater proportion of affirmative responses from problematic respondents compared to those that are more common. The lowest discrepancy between problematic and non-problematic respondents was in using cleaner or disinfectant on one's hands or skin, which is only moderately

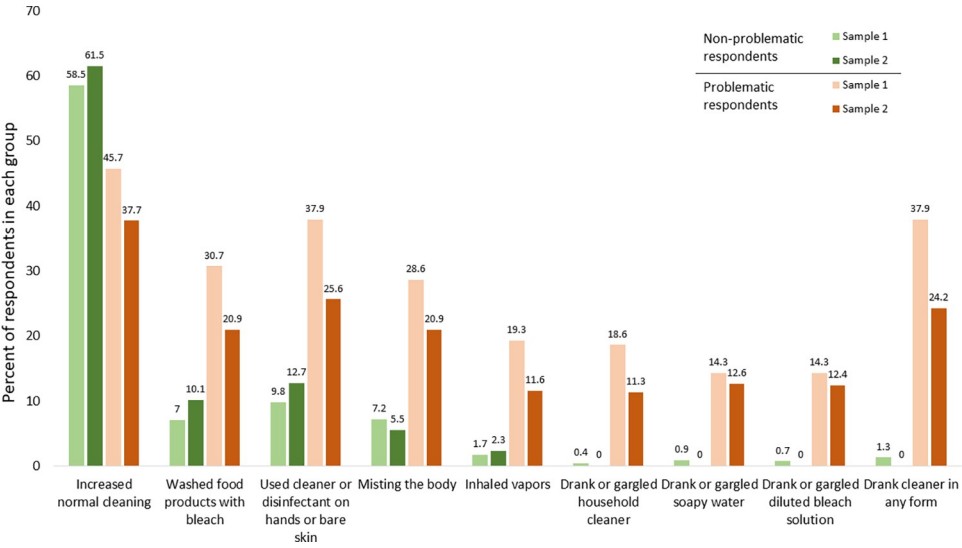

**Fig 1. Comparisons of problematic and non-problematic respondents' reports of cleaning and disinfection practices between April and July, 2020.** *Note*: For the first five behaviors in the graph, the problematic sample was determined by the attentiveness and acquiescence data quality measures for both Sample 1 and Sample 2. For the last three highly dangerous behaviors (drinking or gargling household cleaner, soapy water, or diluted bleach), the sorting into problematic or non-problematic respondents in Sample 2 additionally includes the response verification measures.

dangerous and more commonly engaged in. For this behavior, problematic respondents were approximately three times as likely as non-problematic respondents to respond affirmatively (Sample 1: 37.9% vs. 9.8%; Sample 2: 25.6 vs. 12.7%). On the other hand, the highest discrepancy between problematic and non-problematic respondents was observed for the least common and highly dangerous behaviors: household cleaner ingestion (i.e., drinking/gargling household cleaner, soapy water, or diluted bleach). Overall, 37.9% (24.2%) of problematic respondents reported engaging in at least one of these practices vs. only 1.3% (0%) of non-problematic respondents.

## Demographic and open-ended response verification (Sample 2 only)

We expected the response verification method to show that most respondents who reported ingesting cleaning products would either fail to confirm doing so or indicate that they did so unintentionally (H3). Only 12 of the 473 respondents who passed the first two data quality measures reported ingesting cleaning products. Of these 12 respondents, only 3 confirmed intentionally selecting "yes" to the question about ingesting a cleaning product. Of the remaining 3 respondents, 2 reported unintentionally ingesting a cleaning product (and therefore not having done so to avoid COVID-19). These results confirm H3—among non-problematic respondents, only 1 respondent verified intentionally ingesting a cleaning product.

After excluding respondents with bad data quality (N = 43), and those who did not verify their response (N = 11), only one respondent remained. We examined this respondent's demographic information and found several suspicious elements. They reported being 20 years old and having four children, which is unusual, especially in the US. More problematically, they also reported a weight of 1900 pounds, and a height of "100". When asked to elaborate on their ingestion of cleansers to prevent a COVID-19 infection, this participant's response was "YXgy-vuguhih". See Fig 2 for a flow chart of the verification process.

Across the entire sample, the open-ended responses further indicated that at least some respondents misunderstood the questions. For example, one respondent who reported drinking and gargling soapy water indicated on the open-ended response that, "My mother made

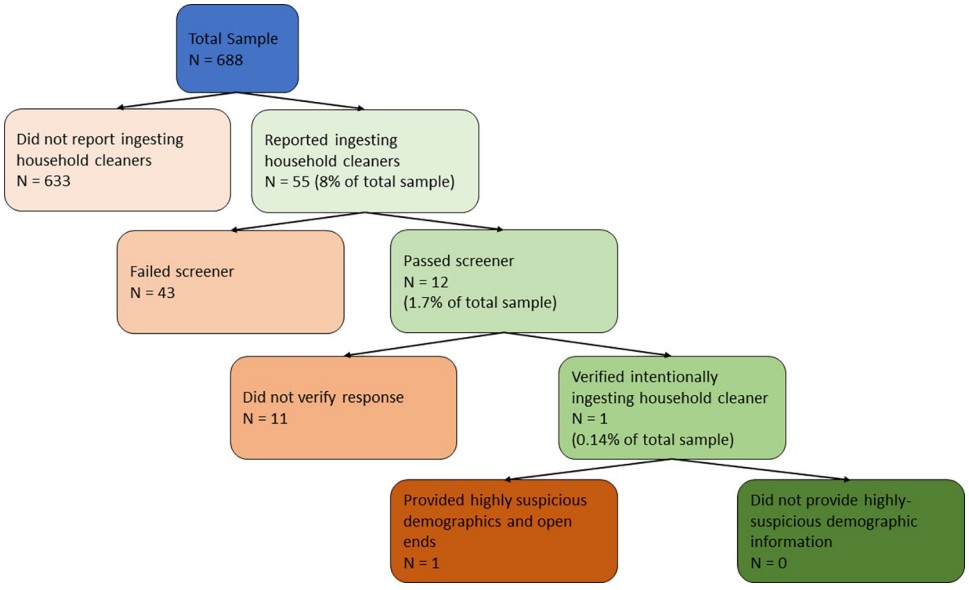

**Fig 2. Flow chart of the stepwise process of verifying the ingestion of cleaning products in Sample 2.**

me wash my mouth out with soap water because I was cursing, and I accidentally swallowed some". It is clear from this response that the respondent did not read the question carefully and did not consider that (a) the question specifically inquired about practices in the time of COVID-19, and (b) specifically inquired about cleanliness practices that were motivated by reducing the likelihood of COVID-19 infection, rather than something that happened to them in childhood.

Overall, we did not find a single respondent who provided any reasonable or compelling open-ended descriptions of ingestion of cleaning products. This is in contrast with other cleaning practices reported on the survey. For example, regarding "misting the body with cleaning spray or alcohol spray after being in public spaces", the open-ended responses were clear and detailed, e.g. "I use 70% alcohol in a spray mist bottle when I take my clothes off and I mist them, let them air dry and then put them in the washing machine." These results confirm H4: there were no cogent responses indicating ingestion of cleaning products to prevent a COVID-19 infection, whereas there were clear and informative reports of other less dangerous cleaning practices.

The open-ended responses indicate that many of the questions were at times misinterpreted even by non-problematic participants. This raises the possibility that the estimates obtained for the non-ingestion practices are also inflated among non-problematic respondents. For example, the meaning of the term "household cleaner" was not clear to all respondents, and at least some put regular soap in that category. To the question "Did you use household cleaner to clean or disinfect bare hands or skin?", examples of open-ended responses included: "I washed my hands with antibacterial soap".

For inhaling vapors of household cleaners, there was also evidence of misinterpretation. Vapors from a wide variety of household products, including cleaners, can be inhaled. The question in the CDC study [24] was specifically intended to detect whether people are inhaling cleaners for the purpose of preventing infection. To achieve this goal, vapors of household cleaner would presumably be inhaled by breathing directly from an open-air container, but the current question does not specify the specific method of delivery. This led some respondents to interpret the question to mean any inhalation of cleaner, even if that cleaner was inhaled during a normal cleaning routine. For example, in response to the question "Did you inhale the vapors of household cleaners like bleach?" examples of open-ended responses included: "I poured product on the floor and began to mop". While inhaling vapors of cleaning products during daily cleaning activities such as mopping can have negative long-term health consequences, such cleaning practices are not considered to pose an acute danger to health.

Other questions also showed evidence of misinterpretation. To the question "Did you mist the body with cleaning spray or alcohol spray after being in public spaces?", examples of open-ended responses included: "I used 70% alcohol in a spray mist bottle when I take my clothes off and I mist them, let them air dry and then put them in the washing machine". This respondent did mist with alcohol, but did not do so directly on the body, and then washed off the alcohol in the washing machine before their clothes made direct contact with the body. To the question of "Why did you wash fruits, vegetables, or other food products with bleach?" examples of misinterpretation included "I used soap and a special sponge to make sure I get a deep clean on the food I serve to my son". As many open-ended responses show, people often generalize to cleaning products other than the specific one that is mentioned in the question.

## Acquiescence bias

We found that 11.9% of respondents reported knowing what the word "wuttlet" means, despite it being a fake word. Further, 5.8% of respondents reported having "suffered a *fatal*

heart attack", and 12.2% claimed to be able to recall from memory the names of every senator who has ever served in the U.S. Senate (these questions were only included in Sample 1).

We created a composite variable for the highly dangerous behaviors by summing the three behaviors. Similarly, we created a composite variable for the three implausible items. Confirming H5, we found a statistically significant, moderate correlation between respondents' reported ingestion of cleaning products and other implausible behaviors ($r = .44$, $p < .001$).

## Dangerous cleaning practices and negative health outcomes

Confirming H6, we found that the association between reported dangerous cleaning practices and negative health outcomes was much higher among problematic respondents compared to non-problematic respondents. Specifically, among problematic respondents in Sample 1 (Sample 2), 27.2% (14.7%) of the variance in the number of reported health symptoms was explained by reported dangerous cleaning practices (Sample 1: $F(1, 138) = 132.3$, $p < 001$; Sample 2: $F(1, 213) = 36.7$, $p < 001$). In contrast, among non-problematic respondents, only 3.5% (0.001%) of the variance in the number of reported health symptoms was explained by reported dangerous cleaning practices (Sample 1: $F(1, 458) = 17.6$, $p < 001$; Sample 2: $F(1, 398) = 0.556$, $p = .82$).

## Discussion

The goal of the present study was to determine whether claims about dangerous cleaning practices to protect against COVID-19 are largely due to problematic respondent bias. Across two studies with nearly 1300 respondents, we replicated the CDC's findings [24] showing that around 4% of respondents reported engaging in each of the three highly dangerous behaviors: drinking or gargling household cleaner, soapy water, and diluted bleach. However, we also observed that 3–7% of respondents reported having never used the Internet while taking the survey online and having suffered a *fatal* heart attack. These findings are consistent with a recent Pew Research Center report that 7% of respondents from over 50 different opt-in panels provide "bogus" data [25], as well as the "lizardman's constant" argument [59], that approximately 4% of survey respondents can be expected to provide nonsense responses to any question.

After categorizing respondents as problematic and non-problematic based on inattention and acquiescence, we found that all reports of highly dangerous cleaning practices came from problematic respondents. Once inattentive, acquiescent, and careless responses were removed from the sample, we find no evidence that people intentionally ingested household cleaning products for protection against COVID-19.

### Types of problematic respondent bias and their effects

We observed that problematic respondent bias introduces two sources of error; 1) random responding increases noise and 2) acquiescence introduces systematic bias.

**Random responding.** Problematic respondents may randomly select from available response options, decreasing the signal-to-noise ratio and driving estimates toward the mean of the distribution. This not only makes rare practices appear more common, but also makes more common practices appear less common. Most non-problematic respondents reported increasing general non-dangerous cleaning practices to prevent a COVID-19 infection, but problematic respondents were less likely to report engaging in such practices. This is consistent with the idea that additional noise attenuates estimates toward the middle of the distribution.

Bias is also proportionally greater among the lowest frequency events. Problematic respondents were two to three times more likely to report using cleaning products on hands or bare skin compared to non-problematic respondents. However, they were eight to twenty-nine times more likely to report gargling or drinking bleach solution compared to non-problematic respondents. Across all cleaning practices examined, the more common practices were proportionally less likely to be affected by problematic respondent bias.

**Acquiescence bias.**   Some problematic respondents systematically select a "Yes" response from among the available response options, introducing error that is correlated across unrelated items. Evidence of this can be seen by examining the correlation between cleaning practices and implausible behaviors and between cleaning practices and negative health outcomes. Among problematic respondents, over 25% of variance in health outcomes is explained by dangerous cleaning practices, but this relationship was not significant among non-problematic respondents. This shows that problematic respondents systematically answer "yes" to a variety of questions across the survey, artificially increasing associations between unrelated events. These results are consistent with previous studies showing that associations between variables are reduced or eliminated when problematic respondents are removed from a sample [18].

## Which dangerous cleanliness practices did people actually engage in?

The spread of COVID-19 created fear of contagion, leading people to seek ways of protecting themselves against infection. The current study explored which specific cleanliness practices people started using during the COVID-19 pandemic, asking respondents about seven practices that are considered dangerous and are not recommended by the CDC. Overall, we found that the reported rates of all seven cleanliness practices are dramatically lower than previously thought because most reports are provided by problematic respondents. The three practices that involve ingesting household cleaning products are entirely due to problematic respondent bias, misreading questions, and respondent error. At the same time, more than 5% of non-problematic respondents reported engaging in several dangerous cleaning practices, including washing food products with bleach, using cleaner or disinfectant on hands and bare skin, and misting the body with cleaning products. However, the interpretation of these data is complicated by open-ended responses showing that some non-problematic respondents did not fully understand the questions. Even attentive and well-intentioned survey-takers can misunderstand questions or may not be aware of the meaning of specific terms and phrases.

Overall, open-ended responses show that some people did not realize the importance of bleach as a specific cleaner and were reporting on using cleaners like soap on their hands and skin or to clean fruits and vegetables. On other questions, respondents were affirming engaging in certain practices, like misting alcohol, but did not realize the importance of other parts of the question, which focused on direct contact with the skin. Additionally, respondents did not always distinguish between practices that they were engaged in prior to the start of the pandemic and those that they started specifically to prevent a COVID-19 infection.

Given the uncertainties in how respondents understood the questions, the implications of these reported practices for public health remain unclear. It is not clear whether substantial numbers of people engage in specific dangerous practices. To fully understand the implication of these cleaning practices for public health, future studies should examine these practices with more specificity, focusing on several details not addressed here or in previous studies. In particular, they should make an effort to define the specific activities and substances in the question to make it clear to respondents which activities they are asking about and define terms such as "household cleaner" to leave no room for doubt that the practices in question pose a health risk. Because the current study and previous studies that used these questions were not

designed to provide a systematic examination of any of these practices, it remains difficult to ascertain whether the practices reported on in this survey were being practiced at all and whether they pose a substantive risk to public health.

## Public health implications

The reasons for the increase in calls to the CDC poison control centers discussed in the introduction are not known. However, even if some of these cases can be attributed to the intentional ingestion of bleach and disinfectants to prevent a COVID-19 infection, it is unlikely to account for the 4% of ingestion of household cleaner reported in the CDC study [24]. Instead, as we show, that 4% ingestion rate is fully accounted for by problematic respondent bias. Our data do not contradict that some individuals may have ingested bleach to prevent a COVID-19 infection, but they do show that such practices are so infrequent that they are not detectable on national surveys.

Health behavior theories emphasize that social norms can impact individual decision making [60] and may be even more salient among vulnerable populations. Experimental evidence suggests that social norms can alter risk perception, thereby influencing behavioral choices [61], and increasing vulnerability to misinformation [62]. Presenting practices such as the ingestion and inhalation of household cleaning products as being practiced by tens of millions of people risks normalizing such practices and potentially inadvertently reinforcing them. For this reason, presenting the results of surveys that are subject to problematic respondent bias is itself a matter of public health concern. The reporting of any rare event detected on a survey should be rigorously examined and should require an additional level of screening.

## Recommendations for eliminating problematic respondents

Across both samples, we revealed high levels of problematic responding (23% and 33% across Samples 1 and 2, respectively). These figures are in line with estimates of problematic responding in the literature, ranging from 7% [25] to 50% [14]. Reports of problematic respondents on online surveys dramatically increased during the first few months of the Covid-19 pandemic. In the Spring of 2020, multiple teams who had collected over 100,000 survey responses on Lucid, reported that rates of problematic responding were between 30% and 84% [63]. This was the same platform from which the CDC had collected their data, and also covers the same time period.

This high level of poor data quality may come as a surprise to many readers, and calls into question many of the reported findings based on data from online opt-in panels. This alarm is well-placed. These findings highlight the importance of using appropriate data quality measures. To accurately measure rare events, researchers should use stricter screening methods when collecting data online. This will improve data quality and increase the signal to noise ratio in any survey. When interpreting survey data that suggests people are engaging in unusual behaviors, researchers should consider the potential influence of problematic respondents on the results. Here, we recommend specific practices that researchers should follow when collecting data online.

**Do not rely on third-party solutions without testing them first.** Standard procedures used in the opt-in panel industry are not sufficient to protect most scientific surveys against bad data, especially when trying to detect rare events. These procedures may protect against duplicate responses, bots, straight lining, and virtual private networks (VPNs), but they are not effective against inattentiveness, mischievous respondents, or acquiescence bias. No solution is perfect, and even if a solution works to protect certain types of surveys, it may not work in all cases.

**Use response validation.** To gather more detailed information about reported behaviors, especially for rare events, researchers should follow up with individual respondents. They can do this by asking respondents to describe their behavior in an open-ended format, providing specific examples and context, and explaining their rationale. Researchers could even set up video interviews with select respondents to verify that they are really who they claim to be, that they understand the questions, and that they are accurately reporting their behaviors. Even a few interviews can provide important evidence that such practices are really occurring in the population.

**Use validated instruments.** Researchers should incorporate validated screeners and other forms of data quality measures into their surveys to protect against problematic respondents. Throughout this paper we have described various validated screening mechanisms, and more in-depth discussions of how to develop and incorporate data protection measures have been described elsewhere. We consider a data quality measure to be validated when: (1) it has been tested in a sample of attentive participants and the vast majority of these participants answered correctly, (2) it specifically detects inattention or acquiescence, rather than participants' memory capacity, education level, or cultural knowledge, and (3) it is neither overly stringent nor overly lenient. This last requirement typically requires extensive testing.

## Conclusion

We found that reports of high-risk cleaning practices to prevent a COVID-19 infection are largely an artifact of problematic respondent bias. Over the last several decades our society has become increasingly dependent on survey research, with more than 80% of surveys using online respondents for at least some of the data collection (Kennedy et al., 2020). Problematic survey respondents pose a fundamental challenge to all survey research and threaten the validity of public-health policy. To mitigate against these threats, researchers should rigorously check for problematic respondents, particularly when the survey aims to measure rare events. Using these techniques significantly increases the accuracy of measurement and prevents problematic respondents from invalidating survey results.

## Author Contributions

**Conceptualization:** Leib Litman, Zohn Rosen, Cheskie Rosenzweig, Sarah L. Weinberger-Litman, Aaron J. Moss.

**Data curation:** Leib Litman, Aaron J. Moss.

**Formal analysis:** Leib Litman, Zohn Rosen, Cheskie Rosenzweig, Aaron J. Moss.

**Methodology:** Leib Litman, Zohn Rosen, Cheskie Rosenzweig, Sarah L. Weinberger-Litman, Aaron J. Moss.

**Project administration:** Leib Litman, Jonathan Robinson.

**Resources:** Jonathan Robinson.

**Writing – original draft:** Leib Litman, Zohn Rosen, Cheskie Rosenzweig, Sarah L. Weinberger-Litman, Aaron J. Moss.

**Writing – review & editing:** Leib Litman, Rachel Hartman.

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
