## [Decision Letter · Decision Letter 0]

23 Nov 2022

PONE-D-22-20912Did People Really Drink Bleach to Prevent COVID-19? A Tale of Problematic Respondents and A Guide for Measuring Rare Events in Survey DataPLOS ONE

Dear Dr. Rachel Hartman,

Thank you for submitting your manuscript to PLOS ONE. After careful consideration, we feel that it has merit but does not fully meet PLOS ONE’s publication criteria as it currently stands. Therefore, we invite you to submit a revised version of the manuscript that addresses the points raised during the review process.

We look forward to receiving your revised manuscript.

Kind regards,

Xin Shen

Academic Editor

PLOS ONE

Journal Requirements:

3. Please note that the name of the ethics committee that approved your study seems to be misspelled in the manuscript (InteRreview instead of InteGreview).

Reviewers' comments:

Reviewer's Responses to Questions

**Comments to the Author**

1. Is the manuscript technically sound, and do the data support the conclusions?

Reviewer #1: Partly

Reviewer #2: Yes

2. Has the statistical analysis been performed appropriately and rigorously? 

Reviewer #1: Yes

Reviewer #2: Yes

3. Have the authors made all data underlying the findings in their manuscript fully available?

Reviewer #1: Yes

Reviewer #2: No

4. Is the manuscript presented in an intelligible fashion and written in standard English?

Reviewer #1: Yes

Reviewer #2: Yes

5. Review Comments to the Author

Reviewer #1: I have read this interesting manuscript and it provides a great insight into a major issue with public health surveys. I was wondering why the author(s) opted to split the study into to and report them somewhat separately. The methods sections for each arm of the study could have been merged in a single section with clear descriptions of the different components. Similarly the results could have been presented in a composite table. A composite summary of the results could have better presented the results more than the narrative description. The use of the flow diagram (although not legible) is also a good way to represent how the problematic respondents were isolated in each arms of the study.

Find additional comments in the text.

Reviewer #2: I recommend publication. Researchers and journalists are far too quick to highlight literally incredible findings from surveys which make no sense and are just plain wrong. The authors' very plausible explanation is that there are wayward (problematic) respondents and once their responses are removed the results become realistic. The paper is, at the very least, a cautionary tale.

I have a couple of suggestions for the revised version.

The paper reads as two papers instead of one. The reader reaches a discussion section and then the second study is introduced. There is a second, much longer, discussion section later. Please integrate the two studies, so the paper reads as a conventional research article rather than unexpectedly, as two.

The paper is too long. It could be a short, sharp interesting piece. I got bored by the second discussion section. The discussion in the second Discussion section on the numbers of calls to poison centers should be in the introduction as indicative evidence that COVID probably did not cause people to ingest disinfectants etc. Similarly, the discussion on the responses to the open-ended questions do not belong in the discussion but in the results section or in an appendix.

It is an interesting topic so let's make an easy read.

The estimate of 23% for problematic respondents is very high and over 30% for the second study is even higher. If representative, these estimates undermine the legitimacy of most estimates from survey research. I would be interested to know if the authors consider that these estimates are realistic for survey research, in general, or are far too high for reasons to do with their studies. Some commentary on the proportion of problematic respondents would be appreciated.

The authors need to acknowledge that the issue of people ingesting household bleach was highly politicized, because of Donald Trump's injudicious and widely misinterpreted comments. I think that if a senior politician from the other side of politics made those comments, there wouldn't have been the media meltdown and maybe the CDC would not have written that report which although didn't mention Trump, was in response to the political climate and had an underlying political message. I realize that we live in high politicized times, but the issue is no longer on the political agenda. I think it is important that the political context of the issue should be acknowledged.

6. PLOS authors have the option to publish the peer review history of their article (what does this mean?). If published, this will include your full peer review and any attached files.

Reviewer #1: No

Reviewer #2: No

---

## [Author Response · Author response to Decision Letter 0]

16 Dec 2022

Miscellaneous:

• Fixed all the headings

• Fixed Table 1

• Fixed title page, changed the author order, and changed the title

• Fixed integReview typo

• Added more details about consent

• Rewrote large portions of the paper for clarity and concision

• We reviewed the reference list and did not see any that have been retracted. We added the following citations: 

o 52. Harrison KL, Zane T. Focus on science: Is there science behind that? Bleach therapy. Sci Autism Treat. 2017;14(1):18-24.

o 54. Khadse PA, Murthy P. Accidental deaths from hand sanitizer consumption among persons with alcohol dependence during the COVID-19 lockdown in India: Analysis of media reports. Asian J Psychiatry. 2021;63:102794. doi:10.1016/j.ajp.2021.102794

o 55. Kochgaway L, Nair AG, Mitra A, Bhargava S, Singh M. COVID casualty: Bilateral blindness due to ingestion of spurious sanitizer. Oman J Ophthalmol. 2020;13(3):164-166. doi:10.4103/ojo.OJO_277_2020

o 63. Aronow PM, Kalla J, Orr L, Ternovski J. Evidence of Rising Rates of Inattentiveness on Lucid in 2020. Published online September 13, 2020. doi:10.31235/osf.io/8sbe4

Reviewer #1: 

I have read this interesting manuscript and it provides a great insight into a major issue with public health surveys. 

I was wondering why the author(s) opted to split the study into two and report them somewhat separately. The methods sections for each arm of the study could have been merged in a single section with clear descriptions of the different components. Similarly the results could have been presented in a composite table. A composite summary of the results could have better presented the results more than the narrative description. The use of the flow diagram (although not legible) is also a good way to represent how the problematic respondents were isolated in each arm of the study.

Thanks for the suggestion (also echoed by reviewer 2)—we agree that combining the studies makes for a better read! We have done as you recommended, combining studies 1 and 2 and inserting a table with a composite summary (and comparison to the CDC study). 

Find additional comments in the text.

These comments were super helpful, thank you! I believe we addressed all of them in the revised text. 

We briefly summarize the changes here:

1. Added a citation for “Despite the widespread use of survey research, self-report data has come under increasing scrutiny over the last ten years due to data quality concerns.”

2. Fixed typo on page 3 (extra word)

3. On page 6, you noted that we didn’t spell out the CDC. However, this was not the first mention (see page 3). 

4. Fixed some of the tenses to past tense where appropriate

5. Fixed typo on page 8

6. Re-did the demographics table to more clearly convey the information

7. Rephrased the sentence on page 11 

8. Relabeled the hypotheses as H1, H2, etc.

9. We added the word “moderate” to accompany the .44 correlation on page 14. Based on Cohen’s conventions, correlations between .3 and .5 are considered moderate. 

10. We removed the rhetorical question on page 24. 

11. We added a reference to “There have been several documented cases of blindness and death due to drinking hand sanitizer since the start of the Covid-19 pandemic”

12. We recreated Figure 3 (now Figure 2).

Reviewer #2: 

I recommend publication. Researchers and journalists are far too quick to highlight literally incredible findings from surveys which make no sense and are just plain wrong. The authors' very plausible explanation is that there are wayward (problematic) respondents and once their responses are removed the results become realistic. The paper is, at the very least, a cautionary tale.

I have a couple of suggestions for the revised version.

The paper reads as two papers instead of one. The reader reaches a discussion section and then the second study is introduced. There is a second, much longer, discussion section later. Please integrate the two studies, so the paper reads as a conventional research article rather than unexpectedly, as two.

Thanks for this great suggestion (also mentioned by Reviewer 1). We combined the two studies. 

The paper is too long. It could be a short, sharp interesting piece. I got bored by the second discussion section. 

We have shortened the paper considerably. 

The discussion in the second Discussion section on the numbers of calls to poison centers should be in the introduction as indicative evidence that COVID probably did not cause people to ingest disinfectants etc. 

We agree with this suggestion, and have moved the relevant paragraphs to the introduction. 

Similarly, the discussion on the responses to the open-ended questions do not belong in the discussion but in the results section or in an appendix.

It is an interesting topic so let's make an easy read.

Good point. We moved the discussion to the results section. 

The estimate of 23% for problematic respondents is very high and over 30% for the second study is even higher. If representative, these estimates undermine the legitimacy of most estimates from survey research. I would be interested to know if the authors consider that these estimates are realistic for survey research, in general, or are far too high for reasons to do with their studies. Some commentary on the proportion of problematic respondents would be appreciated.

We commented on this in the introduction, but have also added a section in the discussion. Here are the two sections:

Introduction:

Problematic respondent bias is a ubiquitous problem that requires mitigation in any type of survey20,28,28, regardless of the modality or demographic population. Thus, the inclusion of rigorous methodology to support the validity of estimates drawn from survey data is critical 37. To this end, researchers have developed data validity screening instruments to combat problematic responses. These instruments can be added before the survey to prevent problematic respondents from participating28, or they can appear within the survey to identify problematic respondents to be excluded from the analytic sample15,29–34. Using such instruments has helped reveal that problematic respondents can drastically attenuate results, at times leading researchers to conclude that previously established findings lack validity17,35,36. This is especially important for studies with direct implications for public health and public policy13.

One popular modality of collecting survey responses is via online opt-in panels, which constitute more that 80% of currently conducted public opinion polls25, and are increasingly used in public health, political science, and social and behavioral sciences28,38. A large literature on opt-in panels indicates that the percentage of problematic respondents on such panels is substantial6,14,25,28,39–44. Estimates of the magnitude of problematic respondent bias in online opt-in panel platforms vary between 4-7%25 and 30%28, although in some studies the magnitude of inattention has been as high as 50%14.

Discussion:

Across both samples, we revealed high levels of problematic responding (23% and 33% across Samples 1 and 2, respectively). These figures are in line with estimates of problematic responding in the literature, ranging from 7%25 to 50%14. Reports of problematic respondents on online surveys dramatically increased during the first few months of the Covid-19 pandemic. In the Spring of 2020, multiple teams who had collected over 100,000 survey responses on Lucid, reported that rates of problematic responding were between 30% and 84%63. This was the same platform from which the CDC had collected their data, and also covers the same time period. 

The authors need to acknowledge that the issue of people ingesting household bleach was highly politicized, because of Donald Trump's injudicious and widely misinterpreted comments. I think that if a senior politician from the other side of politics made those comments, there wouldn't have been the media meltdown and maybe the CDC would not have written that report which although didn't mention Trump, was in response to the political climate and had an underlying political message. I realize that we live in high politicized times, but the issue is no longer on the political agenda. I think it is important that the political context of the issue should be acknowledged.

We agree that the toxic political environment likely contributed to the existence of the CDC report. We think there is a cautionary tale to be told, for both scientists and journalists, about the eagerness to accept and promote findings that align with one’s political ideology. We hope that readers of our paper will take note of these implications. However, we’re reluctant to comment on the political aspect of this issue within the paper, because to some readers this might make us appear politically biased or not impartial, and it could lower our credibility. For the paper to remain as objective as possible, we’re happy for people to draw their own conclusion. We think after our paper is published, there’s room for other papers (e.g., op eds) to be published, that expand on the role of politics. We’d be happy to collaborate with anyone interested in pursuing that angle in alternative avenues, but this paper doesn’t seem like the best place for it.

---

## [Decision Letter · Decision Letter 1]

14 Jun 2023

Did people really drink bleach to prevent COVID-19? A guide for protecting survey data against problematic respondents.

PONE-D-22-20912R1

Dear Dr. Hartman,

We’re pleased to inform you that your manuscript has been judged scientifically suitable for publication and will be formally accepted for publication once it meets all outstanding technical requirements.

Kind regards,

Xin Shen

Academic Editor

PLOS ONE

Additional Editor Comments (optional):

Dear authors,

In order to expedite the publication of the qualified manuscript, I checked your response to the one of last reviewers who had no response. I noticed your response to the reviewer and  you have addressed all concerns.

Please note that I have acted as a reviewer for this manuscript, and you will find my comments below, under Reviewer #3.

For the expedition of publication, please follow our journal style and double-check the format.

Best wishes,

Xin Shen

Reviewers' comments:

Reviewer's Responses to Questions

**Comments to the Author**

1. If the authors have adequately addressed your comments raised in a previous round of review and you feel that this manuscript is now acceptable for publication, you may indicate that here to bypass the “Comments to the Author” section, enter your conflict of interest statement in the “Confidential to Editor” section, and submit your "Accept" recommendation.

Reviewer #2: All comments have been addressed

Reviewer #3: All comments have been addressed

2. Is the manuscript technically sound, and do the data support the conclusions?

Reviewer #2: Yes

Reviewer #3: Yes

3. Has the statistical analysis been performed appropriately and rigorously? 

Reviewer #2: Yes

Reviewer #3: Yes

4. Have the authors made all data underlying the findings in their manuscript fully available?

Reviewer #2: Yes

Reviewer #3: Yes

5. Is the manuscript presented in an intelligible fashion and written in standard English?

Reviewer #2: Yes

Reviewer #3: Yes

6. Review Comments to the Author

Reviewer #2: Thet have addressed my concerns and I am happy to recommend publication. The authors should check and recheck the text.

Reviewer #3: (No Response)

7. PLOS authors have the option to publish the peer review history of their article (what does this mean?). If published, this will include your full peer review and any attached files.

Reviewer #2: **Yes: **Gary N. Marks

Reviewer #3: No
